# Effect of PEG Anchor and Serum on Lipid Nanoparticles: Development of a Nanoparticles Tracking Method

**DOI:** 10.3390/pharmaceutics15020597

**Published:** 2023-02-10

**Authors:** Manon Berger, Manon Degey, Jeanne Leblond Chain, Erik Maquoi, Brigitte Evrard, Anna Lechanteur, Géraldine Piel

**Affiliations:** 1Laboratory of Pharmaceutical Technology and Biopharmacy, CIRM, University of Liege, 4000 Liège, Belgium; 2University of Bordeaux, CNRS, INSERM, ARNA, UMR 5320, U1212, F-33000 Bordeaux, France; 3Laboratory of Tumor and Development Biology, GIGA-Cancer, University of Liege, 4000 Liège, Belgium

**Keywords:** NTA analysis, LNP, protein corona, lipid-PEG

## Abstract

Polyethylene glycol (PEG) is used in Lipid Nanoparticles (LNPs) formulations to confer stealth properties and is traditionally anchored in membranes by a lipid moiety whose length significantly impacts the LNPs fate in vivo. C18 acyl chains are efficiently anchored in the membrane, while shorter C14 lipids are quickly desorbed and replaced by a protein corona responsible for the completely different fate of LNPs. In this context, a method to predict the biological behavior of LNPs depending on the lipid-PEG dissociation was developed using the Nanoparticle Tracking Analysis (NTA) method in serum. Two formulations of siRNA-containing LNPs were prepared including CSL3 or SM-102 lipids and were grafted with different lipids-PEG (C18, C14 lipids-PEG, and Ceramide-PEG). The impact of the lipid-PEG on the interactions between LNPs and serum components was demonstrated by monitoring the mean particle size and the concentration over time. In vitro, these formulations demonstrated low toxicity and efficient gene knockdown on tumor MDA-MB-231 cells, but serum was found to significantly impact the efficiency of C18-PEG-based LNPs, while it did not impact the efficiency of C14-PEG-based LNPs. The NTA method demonstrated the ability to discriminate between the behaviors of LNPs according to serum proteins’ interactions. CSL3 lipid and Cer-PEG were confirmed to have promise for LNP formulation.

## 1. Introduction

The recent emergence of COVID-19 vaccines composed of Lipid Nanoparticles (LNPs) and mRNA has highlighted the wide range of diseases that could be treated with lipid nanovectors. LNPs traditionally include nucleic acids (mRNA, siRNA) and four types of lipids: an ionizable lipid, cholesterol, a phospholipid, and a lipid bound to poly(ethylene glycol) (lipid-PEG) [1]. The lipid-PEG is known to play a major role in the outcome of vectors, and it should be selected according to the application. Indeed, both the lipid and the PEG portions have an important impact on the in vivo behavior of the LNPs. While the use of PEG is important in order to confer steric stability and increase the elimination half-time of particles, the type of lipid associated with the PEG can therefore have a significant impact on its in vivo behavior. It is indeed largely described that one of the most important limitation between in vivo and in vitro translation is the important changes to LNPs’ identities when they reach biological fluids [2]. Regarding the PEG chain, an intermediate length—such as molecular weight of 2000 g/mol, i.e., ~45 repetition units (PEG_2000_)—is commonly used, as this provides a good compromise between increased elimination half-time and efficient gene delivery. Short PEGs (PEG_1000_ or shorter) do not prevent protein corona formation and fail to increase the particles’ blood circulation time, whereas long PEGs (PEG_5000_ or longer) can strongly interfere with the cellular uptake or the endosomal escape process [3,4,5]. Moreover, the lipid anchoring the PEG also demonstrated a significant impact on nanovector’s outcome. Previous studies have shown that PEGs with short lipid chains such as the DMG-PEG_2000_ (C14, dimyristoyl-glycerol) rapidly dissociate from the lipid membrane of LNPs in serum, thereby reducing their circulation time compared to LNPs composed of longer lipid-chains, such as DSPE-PEG_2000_ (C18, distearoyl-phosphatidylethanolamine) [6]. A few hours following parenteral injection, DMG-PEG_2000_ is dissociated and replaced by a biomolecular corona including apolipoprotein E (ApoE) with specificity for hepatocytes [7,8]. DMG-PEG_2000_ is therefore optimal for liver targeting, and the development of Onpattro^TM^ is a concrete therapeutic application approved by the FDA. In this context, two short lipids-PEG (C14) were recently used in the Comirnaty^®^ and Spikevax^®^ COVID-19 vaccines: ALC-0159 (DTA-PEG_2000_, ditetradecylacetamide) and DMG-PEG_2000_, respectively. After an intramuscular injection, LNPs are responsible for a transient local inflammation and the recruitment of neutrophils and antigen-presenting cells [1]. In contrast, longer PEG-derivatives, such as C18 lipids-PEG, display a better anchoring ability in lipid membranes. For example, DSPE-PEG_2000_ is used to effectively protect LNPs by decreasing interactions with blood proteins, allowing for a prolonged blood circulation time and increased tumor accumulation [3,9].

In this context, the objective of this study was to develop an in vitro method to rapidly assess the behavior of LNPs in serum in order to predict their in vivo fate. For this purpose, two LNPs formulations [CSL3 or SM-102 (pH-sensitive lipids)/DSPC/Chol/lipid-PEG in 50/10/37.5/2.5 molar ratio] carrying siRNA were produced. The ionizable lipids were selected based on the promise they show for efficiently delivering nucleic acids. The SM-102 lipid (Figure 1B) was recently used in the Spikevax^®^ COVID-19 vaccine. It has a tertiary amine function allowing for electrostatic complexation with nucleic acids at acidic pH and then LNPs’ stability at physiological pH [1,10]. The switchable cationic lipid CSL3 (Figure 1B) has previously demonstrated its ability to promote endosomal escape and deliver siRNA through a pH-triggered conformational switch [11,12,13]. For both formulations, five different lipids-PEG were used (Figure 1A): two C18 lipids (DSPE- and DSG-PEG), two C14 lipids (DMG- and DTA-PEG [ALC-0159]) and one asymmetric ceramide lipid (C16 and C8 [Cer-PEG]) with intermediary lipid length. LNPs were produced by a rapid-mixing method. The commercial SM-102 lipid was compared with the recently developed CSL3 lipid.

In order to study the interactions between LNPs and serum components, a Nanoparticle Tracking Analysis (NTA) method which allows for the individual tracking of particles and the determination of the particle concentration [14,15] has been developed. LNPs incubated in serum were analyzed in terms of mean size and particle concentration over time. Indeed, the inability of the dynamic light scattering (DLS) method to analyze polydisperse samples and the important contribution of large particles in the intensity distribution profile obtained using this method [14,16] are the major drawbacks of DLS analysis, both of which make the method inaccurate for evaluating the formation of the protein corona. Moreover, compared with traditional particles characterization techniques, NTA allows for the combination of laser light scattering microscopy with a camera in order to visualize the particles in real-time and to record them [14]. The impact of these lipids-PEG and of the ionizable lipids on the cytotoxicity, transfection, and gene delivery efficacy of LNPs in tumor cells was also studied in serum-containing medium using live imaging. This study will allow for the development of an experimental method which could speed-up the translation of the multiple new LNPs in development from in vitro to in vivo.

## 2. Materials and Methods

### 2.1. Material

Cholesterol (Chol), 1,2-distearoyl-sn-glycero-3-phosphocholine (DSPC), 1,2-distearoyl-sn-glycero-3-phosphoethanolamine-N-[methoxy(polyethylene glycol)-2000] (DSPE-PEG), distearoyl-rac-glycerol-PEG2000 (DSG-PEG) and N-octanoyl-sphingosine-1-{succinyl[methoxy(polyethylene glycol)2000]} (Cer-PEG) were purchased from Avanti Polar Lipids, Inc. (Alabaster, AL, USA); 1-octylnonyl 8-[(2-hydroxyethyl)[6-oxo-6-(undecyloxy)hexyl]amino]-octanoate (SM-102), dimyristoyl-glycerol-methoxypolyethylene glycol-2000 (DMG-PEG) and ditetradecylacetamide-methoxypolyethylene glycol-2000 (DTA-PEG) were purchased from SINOPEG (Xiamen SINOPEG Biotech Co., Ltd, Fujian, China). CSL3 lipid was provided by Dr. Jeanne Leblond Chain (University of Bordeaux, France) and was custom synthesized by Richman Chemicals (Lower Gwynedd, PA, USA) according to the previously described procedure [11]. A Quant-iT™ RiboGreen^®^ RNA assay was obtained from Invitrogen^TM^ (ThermoFisher Scientific, Walthman, MA, USA). RNAse-free water (Nuclease-Free, DEPC-treated) was purchased from Fisher bioreagents (ThermoFisher Scientific, Walthman, MA, USA). Phosphate-buffered saline (PBS) tablet and sodium acetate trihydrate were purchased from Sigma-Aldrich (Merck KGaA, Darmstadt, Germany).

HepG2 and MDA-MB-231 cell lines were obtained from the American Type Culture Collection (ATCC, University Blvd, Manassas, VA, USA), and HeLa cells were provided by Dr. Jeanne Leblond (University of Bordeaux, France). Dulbecco’s Modified Eagle Medium (DMEM): High Glucose was purchased from Biowest (Nuaillé, France). Heat-inactivated fetal bovine serum (FBS), penicillin streptomycin (PenStrep^®^), RPMI 1640 medium and Opti-MEM^TM^ were purchased from Gibco^TM^ (ThermoFisher Scientific, Walthman, MA, USA). FBS was purchased from Sigma-Aldrich. Lipofectamine^®^ RNAiMAX was obtained from ThermoFisher Scientific (Walthman, MA, USA).

siRNA active against EGFP (siGFP) and negative control siRNA GL3 (siGL3) were provided by Eurogentec^®^ (Eurogentec SA, Liège, Belgium) with the following sequences: siGFP: sense strand: 5′-GCAAGCUGACCCUGAAGUUC55-3′, antisense strand: 5′-GAACUUCAGGGUCAGCUUGC55-3′; siGL3: sense strand: 5′-CUUACGCUGAGUACUUCGAUU55-3′, antisense strand: 5′-AAUCGAAGUACUCAGCGUAAG55-3′.

### 2.2. Preparation of Lipid Nanoparticles (LNPs)

The LNPs were prepared by a rapid-mixing method as previously described [11,17,18]. SM-102 or CSL3, DSPC, Chol and lipid-PEG (DSPE-, DSG-, DTA-, DMG- or Cer-PEG) dissolved in ethanol (1 mM total lipid concentration) in a molar ratio of 50/10/37.5/2.5 (0.5 mL) were mixed with an siRNA solution (1.5 mL) prepared in 25 mM of acetate buffer at pH 4 (SM-102 formulations) or prepared in PBS buffer pH 7.4 (CSL3 formulations) at an N/P ratio of 4. Mixing occurred following a rapid-mixing home-made method which used two syringe pumps (Chemyx Fusion 200-X, KR Analytical Ltd., Sandbach, UK) connected by a T-junction (PEEK™ Tee, ThermoFisher Scientific [Walthman, MA, USA]) and Tube Peek (1/16″, 0.010″ between the pump and the T-junction, 1/16″ 0.020″ after the T-junction). Mixing occurred at a total flow rate of 12 mL/min with a flow rate ratio of 3:1 (aqueous to ethanol). The LNPs were then dialyzed overnight against PBS buffer at pH 7.4, at 4 °C, and under magnetic stirring. Pur-A-Lyzer^TM^ Maxi dialysis tubes MWCO 12–14 kDa (Sigma-Aldrich, St. Louis, MO, USA) were used.

### 2.3. LNPs Characterization

#### 2.3.1. Size, PdI, Surface Charge and Concentration

The LNPs in RNAse-free water (5 times dilution) were sized in terms of Z-average size (nm) using dynamic light scattering (DLS) at 25 °C with a fixed angle of 90° using the Malvern Zetasizer^®^ (Nano ZS, Malvern Instrument, Worchestershire, UK). The polydispersity index (PdI) and zeta potential (mV) were determined with the same instrument. All of the experiments were measured in triplicate (n = 3) at 25 °C. Mean size and LNPs concentration were also measured by NTA using the NanoSight NS300 (Malvern Instrument, Worchestershire, UK).

#### 2.3.2. Complexation Efficiency

The quantity of the uncomplexed siRNA was assayed using a Quant-iT^TM^ RiboGreen^®^ RNA assay following the manufacturer’s instructions. siRNA calibration curves were prepared as previously described [19], and LNP samples were diluted in RNAse-free water. To each well 100 μL of RiboGreen^®^ reagent was added, and the fluorescence was measured using the FlexStation 3 Multi-Mode Microplate Reader (Molecular Devices, CA, USA). The wavelengths of excitation and emission were 485 nm and 530 nm, respectively. The detected fluorescence was related to the concentration of free siRNA according to the blank and the calibration curves. Next, 50 µL Triton X-100 (10%) was added to each well in order to disrupt all lipid vesicles and release the encapsulated siRNA, and fluorescence was measured again. The encapsulation efficiency (EE) was calculated as follow:$$\mathrm{EE}\;(\%)=(\frac{\left(\left[C2]-[C1\right]\right)}{\left[C2\right]})\times 100,$$
where [C1] is the siRNA concentration without Triton X-100, and [C2] is the siRNA concentration with Triton X-100.

### 2.4. NTA Analysis Method to Study the Protein Corona Formation

The protocol was inspired by Karim et al. [20]. 150 µL of the LNPs were diluted with RNAse-free water at a ratio of 1:2 (*v*/*v*). Next, 150 µL of the formulations was mixed with RNAse-free water and FBS at a ratio of 1:1:1 (*v*/*v*) and incubated for several hours at 37 °C under stirring. The mixtures were diluted at a final volume of 1 mL using RNAse-free water in order to obtain the recommended particle concentration range. Particles size and concentration were measured with the NanoSight NS300. The formation of the protein corona around the LNPs was evaluated following the size distribution and the concentration at the initial time (no FBS) and after different incubation times in FBS (LNPs–FBS mixture) at 25 °C. The samples were measured in triplicate (n = 3) for 60 s at 25 °C using a red laser (642 nm), sCMOS camera and under a constant syringe pump speed of 40 (arbitrary units). Data were analyzed using NTA 3.4 Build 3.4.003 software, with a detection threshold 2 or 3 depending on the sample.

### 2.5. Cell Cultures

Human MDA-MB-231-expressing mEmerald protein, HeLa, and HepG2 cells were used for in vitro tests. MDA-MB-231 cells expressing the mEmerald protein were generated by lentiviral transduction. Gene transfer lentiviral plasmid pLV SV40 mEmerald (Puro) was purchased from E-Zyvec (Villeneuve d’Ascq, France). This plasmid allows for cytoplasmic mEmerald fluorescent protein and selection marker (puromycin) expression driven by the SV40 or PGK promoter, respectively. The lentiviral vector was generated by the GIGA Viral Vectors platform (University of Liège). Briefly, Lenti-X 293T cells (Clontech^®^, 632180) were co-transfected with a pSPAX2 (Addgene^®^, Cambridge, MA, USA) and a VSV-G encoding vector [21]. Viral supernatants were collected at 48 h, 72 h and 96 h post-transfection, filtrated (0.2 µM) and concentrated 100× by ultracentrifugation. The lentiviral vectors were then titrated with a qPCR Lentivirus Titration (Titer) Kit (ABM^®^, LV900, Richmond, BC, Canada). MDA-MB-231 cells were transduced with the lentiviral vector (70 TU/cell). Transduced cells expressing mEmerald were selected with Puromycin (InvivoGen) (1 µg/mL) and fluorescence-activated cell sorting (FACSAria III, GIGA-Flow Cytometry platform, University of Liège). The absence of RCL and mycoplasma in cell supernatant was confirmed using a qPCR Lentivirus Titration kit and MycoAlert™ PLUS Mycoplasma Detection Kit (Lonza, LT07-710), respectively.

Cells were maintained in DMEM (MDA-MB-231 and HeLa) or RPMI (HepG2) media supplemented with 10% FBS and 1% of PenStrep^®^, at 37 °C, in 5% CO_2_-humidified atmosphere.

### 2.6. Evaluation of Metabolic Activity

Cell metabolic activity was determined using a Resazurin reduction assay (Resazurin sodium salt, Sigma-Aldrich, St. Louis, MO, USA). HepG2 and HeLa cells were seeded into 12-well plates at a density of 8 × 10^4^ cells/well 24 h before transfection. After 24 h (60–70% confluence), the culture medium was replaced by 0.5 mL of Opti-MEM^TM^ containing LNPs (100 nM of siRNA GL3) or LNPs supplemented with 10% FBS. Triton X-100 diluted in Opti-MEM^TM^ medium was used as a cytotoxic positive control. Cells without the addition of LNPs were also seeded and were considered as blank cells. Cells were incubated for 4 h at 37 °C, and then medium containing the formulations was replaced by fresh culture medium. After 24 h of incubation, 90 µL of Resazurin was added into each well and incubated at 37 °C for 2 h. Then, 100 μL of each condition was added in triplicate into a 96-well plate and measured with a fluorescent spectrophotometer (Molecular Devices, San Jose, CA, USA). The wavelengths of excitation and of emission were 560 nm and 590 nm, respectively.

### 2.7. Live Cell Imaging

MDA-MB-231 cells expressing cytoplasmic mEmerald protein were seeded into 24-well plates at a density of 3 × 10^4^ cells/well 24 h before transfection. After incubation (60% confluence), culture medium was replaced by 0.3 mL of Opti-MEM^TM^ containing LNPs (100 nM of GFP siRNA) or LNPs supplemented with 10% FBS or heat-inactivated (HI) FBS for 4 h, and the transfection medium was replaced by fresh culture medium. Cell population images were obtained over time using an IncuCyte SX5 live cell imaging system (Sartorius, Göttingen, Germany) residing within an In-Vitro Cell ES NU-5831 (NuAir) tissue culture incubator maintained at 37 °C with 5% CO_2_. The phase contrast and fluorescence images were acquired over time, once before transfection, and then every 6 h for 72 h after transfection. Images were acquired using a 10× objective lens in phase contrast and in green fluorescence channels (ex: 453–485 nm, Em: 494–533 nm). Nine images were acquired from each well every 6 h. Segmentation analysis was performed using the IncuCyte Cell-by-Cell Analysis Software Module (segmentation adjustment: 1, cell detection sensitivity: 0.5, cell contrast: 2, cell morphology: 3, surface fit) and the green fluorescence mean intensity of each segmented cell from the last acquisition was used to create a frequency distribution. For each experiment (n = 3), the fluorescence intensity of each cell measured at 78 h (last acquisition) was used to calculate a population median value for each formulation tested. These median values were then used to quantify the inhibition of fluorescence intensity relative to untreated cells. In order to monitor the evolution of the green fluorescence over time, the mean mEmerald fluorescence intensities (GCU) of each well for each time frame and for the three experiments were plotted as a heatmap.

### 2.8. Statistics

Values were expressed as means ± standard deviations (SD) for at least n = 3. PRISM (GraphPad Prism 9) was used for statistical analysis. Data were compared among groups using ANOVA test followed by the Tukey or Dunnett post-test depending on the method. The difference between groups was considered significant when the *p*-value was <0.05 (*), <0.01 (**), <0.001 (***) or <0.0001 (****).

## 3. Results

### 3.1. Physicochemical Characterization of LNPs

LNPs composed of five different lipids-PEG and two different ionizable lipids were produced using a rapid-mixing method and were characterized in terms of size (Z-average), polydispersity (PdI) and zeta potential using the DLS; in terms of mean size (Mean), SD, and concentration using the NTA; and in terms of siRNA encapsulation using the Quant-iTTM RiboGreen^®^ RNA assay. Excepting the CSL3/Cer-PEG formulations, all of the formulations showed a Z-Average (DLS) between 70 and 195 nm and a Mean (NTA) between 80 and 162 nm (Figure 2A). Surprisingly, by DLS, CSL3/Cer-PEG LNPs showed a larger size (around 300 nm) than other LNPs, while by NTA, the size was similar to the others. Based on the results obtained by the two complementary DLS and NTA methods, several parameters were studied. Regarding the impact of the lipid-PEG on the LNP size, for both methods and both ionizable lipids (CSL3 or SM-102 formulations), the DSPE formulation was taken as a control and compared with the other formulations of the corresponding group. Although larger sizes were measured using DLS for the C14 and C8-Ceramide formulations compared to the C18 formulations (over 170 nm for CSL3 and over 100 nm for SM-102 formulations), no significant correlation between lipid-PEG length and LNP size could be made. The impact of the ionizable lipid was then studied for each method by comparing each CSL3 formulation with the corresponding SM-102 formulation including the same lipid-PEG. Slightly smaller sizes were observed for SM-102 formulations using both DLS and NTA methods, and statistical analysis confirmed that SM-102 formulations demonstrated significantly smaller sizes (Z-average and Mean size) compared to the same CSL3 formulation, except for DSPE-PEG LNPs, as shown in Figure 2A. Finally, the impact of the analytical method used was studied by comparing the data obtained by DLS or NTA for each formulation. A significant difference was observed for the CSL3/Cer-PEG formulation, which can be explained by the very high size measured by DLS, but no significant difference between the methods was observed for the other formulations.

A PdI between 0.037 and 0.466 was measured for all LNPs, but no correlation with the lipid-PEG length could be made when CSL3/DSPE-PEG LNPs were taken as control (Figure 2B). Unsurprisingly, neutral zeta potential was measured for all of formulations, as LNPs contained ionizable lipids with a pKa lower than 7.0.

Finally, no impact of the lipid-PEG was observed on the siRNA encapsulation level when CSL3 and SM-102/DSPE-PEG were taken as control and compared with the other formulations of the corresponding group (Figure 2D). Similar high levels of siRNA encapsulation (~80%) were observed for all of the CSL3 formulations, as previously measured [11] (Figure 2D), but lower encapsulation efficiencies were measured for SM-102 LNPs with ~50% of encapsulated siRNA. Moreover, statistical analysis confirmed that the SM-102 formulations including Cer- and DTA-PEG showed significantly lower levels of siRNA encapsulation when compared with the corresponding CSL3 formulations.

### 3.2. Evaluation of the Protein Corona Formation

The evolution of LNP size distribution profiles after incubation in fetal bovine serum (FBS) was measured by NTA in order to assess the formation of a protein corona around particles. Two main parameters were considered: the average size (Mean) and the concentration of total particles (LNPs at the initial time, LNPs and serum particles during the incubation). As shown in Figure 3, after incubation in FBS, the particle size distribution profiles (concentration/size curves) were completely different depending on the lipid-PEG included in the LNPs. A flattening and displacement of the curves was observed for the DMG- and DTA-PEG (C14)-based formulations (Figure 3G–J), indicating an increase in the mean size and a decrease in the total particle concentration over time in the presence of FBS. The increase in the mean size was related to the adsorption of proteins around LNPs. On the contrary, the DSPE- and DSG-PEG (C18)-based formulations (Figure 3A–D) did not show any flattening of their curves over time. These formulations were characterized by maintaining a constant average size with no decrease in concentration during incubation. The formulations containing Cer-PEG demonstrated intermediate results in terms of size increase and concentration decrease.

Figure 4 summarizes the observations of the total particles mean size (Figure 4A) and concentration (Figure 4B) evolution as a function of time. Formulations containing C14 lipids-PEG (DMG- or DTA-PEG) showed a significant increase in the mean size (from 50 to more than 100 nm) and a decrease in concentration (normalized concentration between 8 and 50% of the initial concentration) after 4 h. Statistical analyses (Figure 4C,D) confirmed that the formulations containing C14 lipids-PEG have a significantly higher mean size after 4 h of incubation in FBS when compared with the CSL3/DSPE-PEG formulation, and a significant decrease in the normalized concentration. These results were related to the emergence of fewer but larger particles following the adsorption of proteins around the LNPs, thereby leading to the alteration of LNPs. Furthermore, differences were observed between DMG- and DTA-PEG formulations. After 4 h of incubation in FBS, a greater increase in size and decrease in concentration was measured for the CSL3/DMG-PEG and SM-102/DMG-PEG formulations as compared to the CSL3/DTA-PEG and SM-102/DTA-PEG formulations, respectively. Also, a greater decrease in concentration was observed for CSL3/DMG-PEG than for CSL3/DTA-PEG. It can therefore be hypothesized that, although both polymers have a similar C14 anchor length, DMG-PEG may desorb more rapidly from the lipid membrane.

In contrast, the DSPE- and DSG-PEG formulations (C18) did not show flattened curves, as presented in Figure 3A–D. These formulations were characterized by the maintenance of a similar average size during incubation (size increase below 12 nm after 4 h of incubation) and the absence of a decrease in concentration. A surprising increase in the concentration was even observed, which could be related to the free serum particles being included in the total count, as they are not adsorbed around the LNPs. This confirmed the good protection against proteins’ interactions provided by C18 lipids-PEG. Intermediate results were obtained for the Cer-PEG formulations. The mean size increase fell between that observed with the C14 lipids-PEG and the C18 lipids-PEG formulations (27 nm and 49 nm for the CSL3 and SM-102/Cer-PEG formulations, respectively, after 4 h). This increase in the mean size was significant for SM-102/Cer-PEG LNPs but not for CSL3/Cer-PEG LNPs. Moreover, a significant decrease in the normalized concentration was observed for both formulations.

Although broadly similar observations can be made for both ionizable lipids (CSL3 and SM-102), differences can also be highlighted. As shown in Figure 4A, the SM-102-based formulations (dotted line) showed a larger size increase than the corresponding CSL3-based formulations (solid line). It can therefore be assumed that the SM-102 formulations provide a lower stability in serum.

Finally, in order to exclude the possibility that the decrease in concentration and increase in size could be related to experimental conditions or to LNPs’ aggregation, LNPs were incubated for 4 h without serum and compared with LNPs incubated in serum. We showed that, without serum, the LNPs’ concentration does not decrease, and the size does not increase, regardless of the lipid-PEG used, meaning that the results observed with DMG-, DTA- and Cer-PEG LNPs are well related to protein corona formation, instead of LNPs aggregation.

### 3.3. Impact on Metabolic Activity in Serum-Containing Medium

In order to evaluate the safety of the tested LNPs, two different human cancer cell lines were used: HeLa and HepG2 cells. HepG2 hepatic cells were used considering the potential hepatic specificity of LNPs covered by a protein corona. Moreover, this cell line is commonly used for hepatotoxicity studies [22]. The metabolic activity study was performed without and with serum (10% FBS) after 24 h of transfection. Data are shown in Figure 5 and were analyzed following several parameters. In serum-free medium (filled bars) and as compared to untreated cells (Dunnett test, Figure 5), the lipid-PEG has no significant impact on the metabolic activity of Hela cells and HepG2 cells, except for SM-102/Cer-PEG LNPs for HepG2 cells. In serum (patterned bars), only one formulation significantly decreased the metabolic activity of HeLa cells (SM-102/Cer-PEG LNPs), while three formulations significantly decreased the metabolic activity of HepG2 cells (SM-102/Cer-, /DMG- and /DTA-PEG LNPs). It was therefore concluded that there was a potential toxicity of C14 lipids-PEG LNPs, mainly in HepG2 cells and in serum. Regarding the impact of the ionizable lipids, the data were compared (Tukey test) between CSL3 and SM-102 formulations including the same lipid-PEG. In serum-free medium, LNPs showed no impact from the ionizable lipid on metabolic activity, while in serum medium, the SM-102/Cer- (*) and /DTA-PEG LNPs (**) showed a significant decrease in metabolic activity compared to the corresponding CSL3/Cer- and /DTA-PEG LNPs, but only for HepG2 cells. These results could also be related to the higher amount of SM-102 LNPs used to transfect cells at a constant concentration of 100 nM siRNA considering that the percentage of siRNA encapsulation was found to be significantly lower for SM-102/Cer- and /DTA-PEG LNPs. Transfection in serum-containing medium seemed to highlight the potential toxicity of the mentioned LNPs.

### 3.4. Impact on Gene Delivery

In order to evaluate the efficacy of LNPs and study the impact of the lipid-PEG and serum on the delivery of a GFP-targeting siRNA, MDA-MB-231 cells expressing mEmerald, a brightly fluorescent monomeric variant of GFP [23], were transfected with LNPs (100 nM siGFP) prepared in Opti-MEM^TM^ supplemented or not with 10% FBS or 10% heat-inactivated (HI) FBS, or Lipofectamine used as a positive control. Cells without the addition of LNPs were used as untreated cells. Single-cell fluorescence was then analyzed by automated live cell imaging. A preliminary image acquisition was made before transfection in order to measure the cells’ basal fluorescence, then the cells were transfected, and a second acquisition was performed after 6 h, which was considered as the first time point after the transfection (LNPs were removed after 4 h) and then every 6 h until 78 h after the first acquisition (Figure 6A). Phase contrast images were used for single cell segmentation and the mean single cell fluorescence intensities measured during the last image acquisition (78 h) were shown as fluorescence frequency distribution plots for each formulation, as shown in Figure 6B (a representative experiment from three independent experiments is shown). The results were analyzed in order to investigate the impact of the lipid-PEG, the ionizable lipid, and the serum on siGFP delivery. Also, for each experiment, the inhibition of fluorescence intensity relative to untreated cells was also analyzed (Figure 7A). First, one-way ANOVA followed by the Dunnett test revealed that all the formulations were able to significantly reduce the mEmerald fluorescence (****) when compared with untreated cells, independently of the serum medium, except SM-102/DSPE-PEG LNPs (ns) when used in HI FBS medium. This significant decrease in fluorescence is illustrated by the shift of the histograms towards low fluorescence values when compared with untreated cells and is consistent with the promising interest of these formulations. Representative microscopy images of the cells from selected experimental conditions are illustrated in Figure 6C. When transfection occurred in serum-free medium, the percentages of inhibition ranged from 72 to 93% for CSL3 LNPs and from 66 to 93% for SM-102 LNPs. As shown in Figure 6B,C, this inhibition was lower for cells treated with DSPE-PEG LNPs (orange histograms) when transfection was performed in serum-containing media (untreated cells and cells treated with SM-102/DSPE-PEG LNPs even displayed nearly identical fluorescence intensity profiles when transfection occurred in HI FBS-containing medium). In contrast, histograms related to cells treated with CSL3- and SM-102/DMG- and Cer-PEG LNPs were similar between the No FBS, FBS and HI FBS conditions. As shown in Figure 7A, the inhibition remained high (from 78 to 93%) for all of the DMG- and Cer-PEG LNPs in serum-containing media, while this inhibition dropped for DSPE-PEG LNPs (about 50% and even 6% for SM-102/DSPE-PEG LNPs when used in HI serum-containing medium). These observations were then related to a possible negative impact of DSPE-PEG on transfection in serum.

In order to study the effect of the lipid-PEG on gene delivery, the DSPE formulations (CSL3 or SM-102 LNPs) were taken as a control and compared to other formulations of corresponding groups (DMG- and Cer-PEG formulations) (Figure 7A). When cells were transfected in serum-free medium, all of the CSL3 LNPs similarly inhibited the mEmerald fluorescence. In contrast, differences were observed for SM-102 LNPs considering that DSPE-PEG LNPs less-efficiently inhibited the fluorescence when compared with Cer-PEG LNPs. In FBS- and HI FBS-containing media, all of the DSPE-PEG LNPs were significantly less inhibitory than the LNPs including C14 lipids-PEG. These results confirmed the hypothesis of a possible negative impact of DSPE-PEG on transfection efficacy in serum-containing media.

When comparing the effects of serum in each formulation, differences in the fluorescence inhibition efficiency were observed. When cells were transfected with DMG- and Cer-PEG LNPs, no significant difference was observed between the No FBS, FBS, and HI FBS conditions, suggesting that the serum has no impact on the transfection efficiency of these C14 lipid-PEG LNPs. As shown in Figure 7A, C14 lipid-PEG LNPs reduced the fluorescence intensity of cells by about 80–90%. In contrast, a significantly lower inhibition was measured when cells were treated with DSPE-PEG LNPs in serum-containing medium as compared to serum-free medium (CSL3/DSPE-PEG: * when transfection occurred in FBS; and SM-102/DSPE-PEG: **** when transfection occurred in HI FBS-containing medium). These results confirmed the hypothesis of the negative impact of DSPE-PEG on the transfection efficacy when performed in serum-containing medium.

The impact of ionizable lipid was then studied for each condition (No FBS, FBS, HI FBS) by comparing each CSL3 formulation with the corresponding SM-102 formulation including the same lipid-PEG. No significant difference was observed, exceptbetween CSL3- and SM-102/DSPE-PEG LNPs, but this was only when cells were treated in HI FBS-containing medium. Nevertheless, this difference might result from the inability of SM-102/DSPE-PEG LNPs to reduce the mEmerald fluorescence intensity as previously mentioned.

Collectively, all of the LNPs tested were able to significantly decrease the fluorescence intensity in MDA-MB-231 cells expressing mEmerald protein in serum-free medium, in a similar way as Lipofectamine, even if this ability was lower for SM-102/DSPE-PEG LNPs. This ability was also significantly reduced when DSPE-PEG LNPs were added to cells in serum-containing medium, highlighting the strong impact of DSPE-PEG on transfection in serum.

The kinetic evolution of the mEmerald fluorescence intensity in MDA-MB-231 cells transfected with the different LNPs in media containing either no FBS, FBS or HI FBS, is shown in Figure 7B. All the tested LNPs reduced the fluorescence intensity, except for SM-102/DSPE-PEG LNPs when used in HI FBS-containing medium, as previously described. This reduction of the fluorescence intensity was found to be similar for Lipofectamine and C14 lipid-PEG LNPs, but not for C18 lipid-PEG LNPs. Interestingly, it was observed that, for most of the tested LNPs, the fluorescence intensity started to decrease from the 2nd acquisition (i.e., from the first image acquisition performed after the end of transfection). A shallower decrease was observed for the DSPE-PEG LNPs, especially in serum-containing medium, but also more generally for SM-102 LNPs. These observations may be related to the fact that C18 LNPs decreased the fluorescence intensity less efficiently, and also to the fact that other efficient LNPs such as SM-102/DMG- or Cer-PEG LNPs may have a delayed effect compared with the corresponding CSL3 LNPs.

## 4. Discussion

As the optimization of LNPs and their translation from in vitro to clinic is complicated by the biological identity of LNPs in contact with biological fluids (phenomenon including the protein corona) [2], particular focus is placed upon the development of experimental techniques for evaluating this phenomenon.

The DLS method is commonly used to characterize nanoparticles, because it allows for fast and easy measurements of the particle [11,24,25,26,27]. In order to study the interactions between nanoparticles and serum components and to conclude on their stability in serum, previous studies used the DLS method [20]. However, bimodal or trimodal size distributions were observed with the control FBS samples and with particles–FBS mixture samples. Indeed, DLS can be used to measure monodisperse samples, but it is inaccurate when analyzing polydisperse samples [14]. Moreover, the intensity distribution leads to an important contribution of large particles [16]. While the DLS is commonly preferred due to its ease of use [14], its use is limited in the case of complex samples. In this context, an alternative method is the NTA, which allows for the visualization and individual tracking of particles. The camera coupled with the laser-illuminated microscope allows to the user to visually confirm changes in size, concentration, etc. Determination of the particle concentration is also one of the main advantages over DLS, as there is no impact of larger particles on the distribution. However, this technique requires more optimization processes and is dependent on user experimentation in regards to the parameter settings [14,15].

Both methods were used to characterize LNPs. The impact of large particles on the intensity distribution profile displayed by the DLS was highlighted when comparing the results obtained with the DLS (Z-average) and the NTA (Mean) methods, as shown in Figure 1A. The mean size given by the NTA was typically lower than the Z-average considering that all of the particles have the same impact on the total particles count. Using this method, all of the formulations displayed a small mean size below 162 nm, but no correlation could be made between the size and the length of the lipid-PEG. The same conclusion can be made for PdI values. If CSL3 formulations demonstrated a significantly higher size compared with SM-102 formulations, CSL3 formulations were able to encapsulate a higher level of siRNA, which should be interesting from the perspective of transfection. Nevertheless, the lipid-PEG showed no impact on siRNA internalization. LNPs showed neutral zeta potential values, which was not surprising considering that they include ionizable lipid with a pKa lower than 7.0 known to exhibit a neutral surface charge at physiological pH [18,28].

As shown in Figure 3 and Figure 4, NTA allowed the discrimination of the different behaviors of LNPs in serum depending on the lipid-PEG included. Particle distribution profiles (Figure 3) were consistent with the rapid dissociation from the lipid membrane of C14 lipids-PEG followed by the adsorption of serum proteins and the alteration of particles. Indeed, flattening of the curves was observed for DMG- and DTA-PEG formulations (C14) over time, and a significant increase in the mean size and decrease in the particle concentrations were observed, meaning important alteration of LNPs when they are not protected anymore by the polymer. On the contrary, the constant concentration-size profiles of DSPE- and DSG-PEG formulations over time were consistent with the theoretical strong anchoring of C18 lipid-PEG and the longer stealth properties they can provide. Regarding the differences observed between DMG- and DTA-PEG LNPs (more important size and concentration changes for DMG-PEG LNPs), it was hypothesized that the difference was related to the chemical structures of these lipids-PEG. Indeed, the DMG segment contains an ester function which could be cleaved more quickly compared to the amide function of the DTA segment (Figure 1A). It was even recently described that the desorption of DMG-PEG from LNPs used in COVID vaccine (Spikevax^®^) is faster than that of DTA-PEG (Comirnaty^®^) [29]. Cer-PEG formulations showed intermediate results for both size and concentration changes over time. The asymmetric structure composed of C8 and C16 acyl chains was identified by the developed method as an intermediate anchor between C18 and C14 lipids-PEG. Indeed, it has already been described that Cer-PEG with short acyl chains (such as C8 chain) exhibits weak anchoring properties when compared with longer acyl chains (as C20 or C24 chain) [30]. In this study, extreme changes in physicochemical properties were observed, and the important increase in the mean size combined with the drastic decrease in concentration for C14 lipids-PEG LNPs can be explained due to the extreme operating conditions that were used to highlight the different LNPs behaviors (33.33% *v*/*v* FBS).

This method was developed using the NTA in order to highlight the dissociation of lipid-PEG in biological fluids. Indeed, DSPE-PEG_2000_ is historically used to increase particles’ lifetime and tumor accumulation [7,31,32,33] due to two parameters. An intermediary PEG chain length (2000 g/mol) provides effective protection against proteins’ adsorption while avoiding a strong decrease in particles cellular uptake or endosomal escape processes [3,4]. Next, the C18 DSPE chain provides an effective anchoring of the PEG into the LNPs, allowing the PEG segment to have a prolonged protection effect with an increased LNPs circulation time. Nevertheless, shorter lipids-PEG have other applications. DMG-PEG has recently been incorporated into the Onpattro^®^ formulation due to its shorter lipid chain anchor, which is rapidly dissociated from the lipid membrane [6,31,33]. When dissociated from their PEG, LNPs can interact with blood proteins, including ApoE proteins, for delivery to the hepatocytes. The use of short lipids-PEG able to quickly dissociate is therefore one of the preferred options to target the liver [34]. In vivo, it was then proved that nanoparticles with DMG-PEG anchor have no prolonged circulation time compared with the same particles grafted with C18-PEG, while the latter demonstrated higher stability in serum and tumor accumulation [33,35]. More recently, DMG-PEG_2000_ was used in the COVID-19 Spikevax^®^ vaccine [36], considering that an extended residence time could lead to undesirable inflammation at the injection site [37], and because, despite the lack of information, opsonization of the LNPs could also help to stimulate the immune system by promoting uptake by innate immune cells [38]. The formation of a protein corona around LNPs can therefore be seen either as a limitation or as an opportunity, depending on the application. In this context, the developed NTA method could be used to quickly screen new LNPs formulations and facilitate the rapid translation from in vitro to clinic. Its ease of implementation (low required volumes and materials) and its benefits over traditional methods to separate LNPs bound to proteins from free proteins make of this method an interesting tool. Indeed, separation methods no longer reflect the practical in vivo dynamic equilibrium between the adsorption and desorption of proteins, or the establishment of a soft corona around the high affinity proteins directly grafted around LNPs forming the hard corona [2].

The impact of these lipids-PEG and of ionizable lipids on cellular metabolic activity was studied in tumor HeLa and HepG2 cells. In serum-free medium, LNPs induced low or no toxicity. In contrast, when used in the presence of serum, a significant impact of SM-102 LNPs with Cer-, DMG- and DTA-PEG on the metabolic activity of cells was noticed, predominantly in HepG2 cells. Considering the previously mentioned rapid desorption of C14 lipid-PEG in serum, it could be hypothesized that the transfection of cells with SM-102 LNPs lacking PEG could disturb cell metabolic activity. Nevertheless, these results should be taken with caution due to the low level of siRNA encapsulation for the SM-102 LNPs, as, in order to work at constant concentrations of siRNA, they required the use of larger volumes and therefore a larger amount of LNPs compared with the CSL3 formulations.

Finally, these LNPs, including siGFP, were tested regarding their efficacy in MDA-MB-231 cells expressing mEmerald fluorescent protein. The impacts of the lipid-PEG, the ionizable lipid and the serum were studied by live imaging over a period of 78 h. The results highlighted the promise of these LNPs, including two recent ionizable lipids (CSL3 and SM-102), as the fluorescence inhibition efficiency achieved in serum-free medium (80–90%) was very similar to that obtained with Lipofectamine, which was used as a positive control. This effect remained very high for C14 DMG- and Cer-PEG LNPs in serum-containing medium (about 80–90%). Nevertheless, that effect was significantly reduced for DSPE-LNPs in serum-containing medium (by about 50%, with no effect for SM-102 LNPs with DSPE-PEG in HI FBS). It was concluded that lipids-PEG with long acyl chains such as C18 could be an obstacle for efficient gene delivery in serum. Indeed, it has recently been described that lipids-PEG with long (C16-C18) alkyl chains could interfere with gene silencing when compared with shorter lipids-PEG, which quickly desorb from membranes [6]. This PEG dissociation was then associated with facilitating the cellular uptake and endosomal escape processes [31]. Recently, the transfection of HepG2 cells with LNPs also showed the impact of C18 lipid-PEG in serum-containing medium. It was observed that the addition of FBS could decrease the transfection efficacy of C18 LNPs while increasing the efficacy of C14 LNPs. The authors discussed these results, suggesting that the protein corona could either hinder the uptake process or facilitate it due to other receptor-mediated phenomenon [4]. Even if the protein corona has been described to inhibit transfection, it now appears that some of the proteins included in this corona could specifically interact with cellular processes and particles trafficking [2]. Regarding the results obtained in this study, it could then be hypothesized that C14 lipids-PEG dissociate in serum, which can facilitate transfection, as PEG is known to prevent the cellular uptake and endosomal escape processes [6]. Following this desorption, PEG could be replaced by a protein corona, which could also play a positive role on transfection, depending on its composition. In contrast, C18 lipids-PEG have strong anchoring ability which allows for the prevention of protein corona formation, but could also be an obstacle to transfection. Moreover, as explained by Chen et al., PEG cannot completely eliminate the binding of proteins around LNPs, and that proteins could impact the transfection in serum.

Several parameters must then be considered when designing LNPs. The use of long lipids-PEG can offer an efficient protection against protein corona formation, prolong the circulation of LNPs, and increase the accumulation in tumor. Nevertheless, the transfection efficacy of these LNPs could finally be lower than expected because of the barrier effect of PEG. However, this effect could be balanced by an intermediate proportion of C18 lipid-PEG in LNPs formulations if an equilibrium between efficient stealth properties and efficient transfection can be found. On the other hand, recent examples highlighted the interest in using short lipids-PEG in order to avoid long effect or to target hepatic diseases. In that case, the formation of the protein corona is intended, and recent studies demonstrated the high transfection efficacy of these LNPs. For further experiments, the CSL3 lipid may be more interesting considering that CSL3 LNPs showed higher siRNA encapsulation levels, lower toxicity and higher efficacy in tumor cells. Moreover, the use of Cer-PEG could be highly interesting due to the high efficacy of Cer-PEG LNPs in serum, and the intermediate results observed in terms of protein corona formation. It could then be a good compromise between stealth properties and gene silencing efficacy.

## 5. Conclusions

Considering the known obstacles that limit the development of therapeutic LNPs as the new identity acquired by these LNPs when they come into contact with biological fluids, a method to predict the biological behavior of LNPs depending on the lipid-PEG dissociation was developed using the powerful nanoparticle tracking analysis (NTA) method in serum. An increase in the mean size and a decrease in the particle concentration over time was correlated with important protein adsorption on LNPs from which C14 lipids-PEG were desorbed. On the contrary, for the C18 lipids-PEG LNPs, the size remained constant and the concentration did not decrease, demonstrating a stronger anchor of C18 lipid-PEG and a longer stealth effect. That method demonstrated the ability to discriminate between the different behaviors of LNPs according to serum proteins’ interactions and could be used to screen new formulations in order to predict their possible fate prior to in vivo testing.

Regarding the tested formulations including promising ionizable lipids (CSL3 and SM-102), the results confirmed, using live imaging, their high transfection efficacy in tumor MDA-MB-231 expressing cytoplasmic mEmerald cells, even in serum medium. Nevertheless, the use of DSPE-PEG in LNPs still requires optimization in order to avoid negative impacts on transfection in serum medium. CSL3 lipid was also confirmed to be promising by demonstrating higher encapsulation ability, lower toxicity and higher efficiency. Finally, Cer-PEG may be used as a compromise in order to confer sufficient stealth properties while not being impacted by serum.

## Figures and Tables

**Figure 1 pharmaceutics-15-00597-f001:**
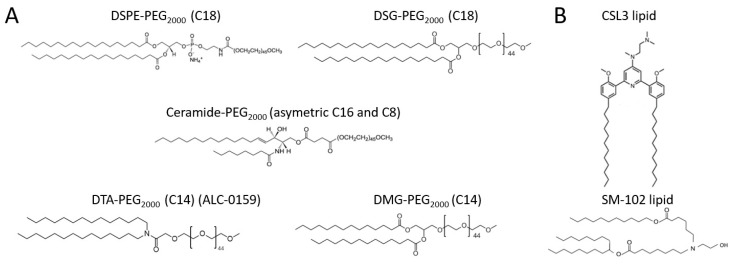
Chemical structures of the (**A**) lipids-PEG and (**B**) ionizable lipids included in the LNPs formulations.

**Figure 2 pharmaceutics-15-00597-f002:**
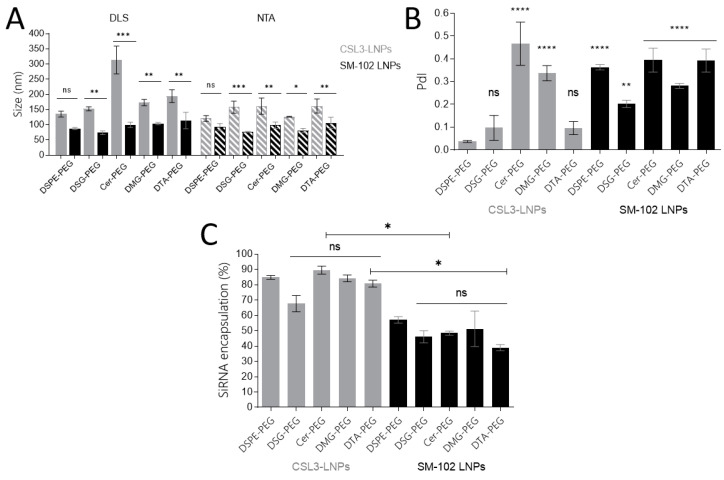
Physicochemical characterization of the CSL3 and SM-102 formulations, including the different lipids-PEG (**A**), Z-average (filled bars) and Mean size (hatched bars) obtained using the DLS and NTA methods, respectively. The impact of the ionizable lipid was studied for each method by comparing each CSL3 formulation (grey) with the corresponding SM-102 formulation (black) including the same lipid-PEG. Statistical analyses compared the CSL3 and SM-102 formulations for each lipid-PEG using the same method. (**B**) PdI obtained by the DLS method. CSL3/DSPE-PEG LNPs were taken as the control. (**C**) siRNA encapsulation efficiency (%). DSPE-PEG formulations were taken as the control and compared with the other formulations of the corresponding group in order to study the impact of the lipid-PEG, and each CSL3 formulation was compared with the corresponding SM-102 formulation including the same lipid-PEG in order to study the impact of the ionizable lipid. Each value represents the mean +/− standard deviation (SD) of three independent experiments (n = 3). Statistical analyses were performed using one-way ANOVA followed by the Tukey test. The difference between groups was considered significant when the *p*-value was <0.05 (*), <0.01 (**), <0.001 (***) or <0.0001 (****).

**Figure 3 pharmaceutics-15-00597-f003:**
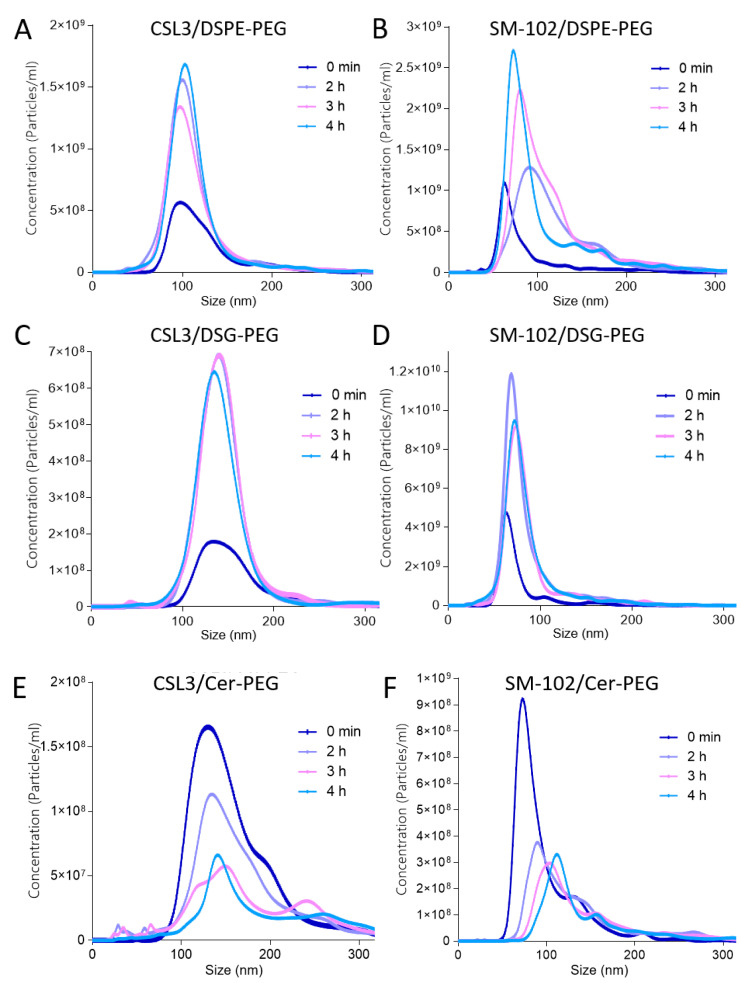
Particles’ distribution profiles (concentration–size curves) given by the NTA during LNPs’ incubation at 37 °C in serum over time (from 0 min to 4 h) (**A**) CSL3/DSPE-PEG LNPs, (**B**) SM-102/DSPE-PEG LNPs, (**C**) CSL3/DSG-PEG LNPs, (**D**) SM-102/DSG-PEG LNPs, (**E**) CSL3/Cer-PEG LNPs, (**F**) SM-102/Cer-PEG LNPs, (**G**) CSL3/DMG-PEG LNPs, (**H**) SM-102/DMG-PEG LNPs, (**I**) CSL3/DTA-PEG LNPs, (**J**) SM-102/DTA-PEG LNPs, (**K**) CSL3/DSPE-PEG LNPs (0 min and 4 h) in water without serum, (**L**) CSL3/Cer-PEG LNPs (0 min and 4 h) in water without serum.

**Figure 4 pharmaceutics-15-00597-f004:**
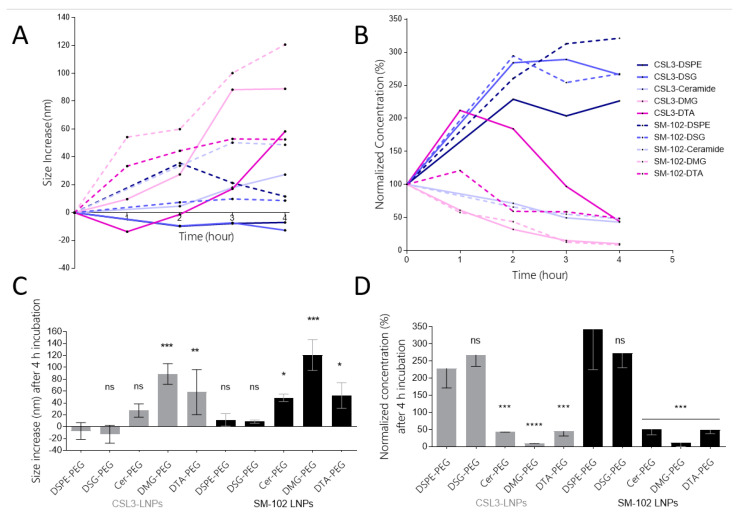
Particles’ (**A**) Mean size increase (nm) and (**B**) normalized concentration (%) over time following incubation in serum. CSL3 formulations are represented by solid lines and SM-102 formulations by dotted lines. Each value represents the mean of three experiments (n = 3), but error bars were voluntarily removed. (**C**) Increase in the mean size and (**D**) decrease in the concentration (normalized to LNPs before the incubation) after 4 h of incubation. Statistical comparisons were performed using one-way ANOVA followed by the Dunnett post-test with CSL3/DSPE-PEG LNPs as the control. The difference between groups was considered significant when the *p*-value was <0.05 (*), <0.01 (**), <0.001 (***) or <0.0001 (****).

**Figure 5 pharmaceutics-15-00597-f005:**
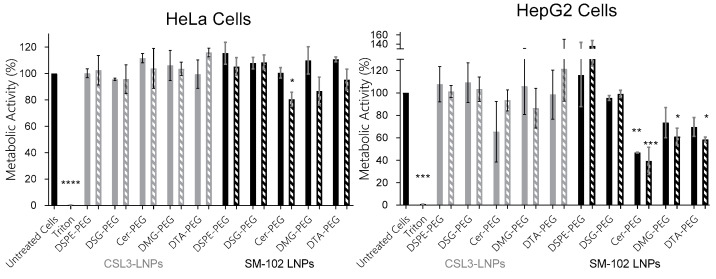
Impact of LNPs on in vitro metabolic activity using Resazurin 24 h post-transfection on two cell lines (HeLa and HepG2 cells). LNPs including GL3 siRNA and prepared in Opti-MEM^TM^ without (filled bars) or with (hatched bars) 10% FBS were added to cells for 4 h. Each value represents the mean +/− standard deviation (SD) of three independent experiments (n = 3). Statistical analyses were performed by one-way ANOVA, followed by the Dunnett post-test with untreated cells as control. The difference between groups was considered significant when the *p*-value was <0.05 (*), <0.01 (**), <0.001 (***) or <0.0001 (****).

**Figure 6 pharmaceutics-15-00597-f006:**
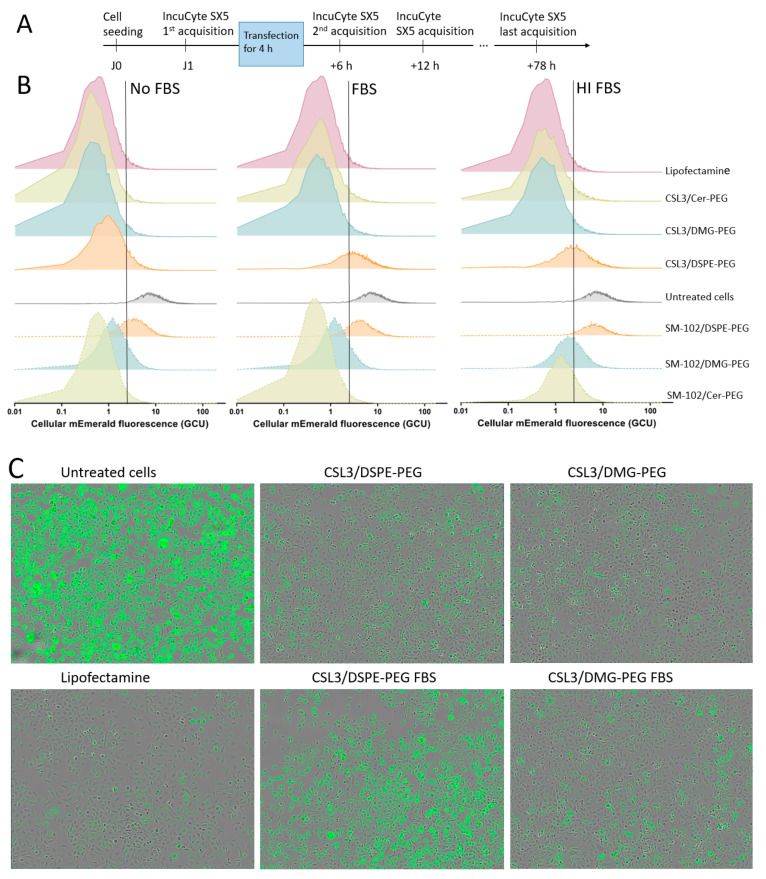
(**A**) Schematic protocol for transfection and image acquisition. (**B**) Frequency distribution of the cellular mEmerald fluorescence intensities (GCU) of MDA-MB-231 cells transfected with CSL3 and SM-102 LNPs, including DSPE-, DMG- and Cer-PEG, in serum-free medium, FBS- or HI FBS-containing media, 72 h after transfection. Untreated- and Lipofectamine-treated cells were used as negative and positive controls, respectively. The vertical lines point to the minimum fluorescence intensity measured in untreated cells. (**C**) Representative composite phase contrast and fluorescent pictures of MDA-MB-231 cells expressing mEmerald and transfected with CSL3 LNPs including DSPE- and DMG-PEG in serum-free or FBS-containing medium, 72 h after transfection.

**Figure 7 pharmaceutics-15-00597-f007:**
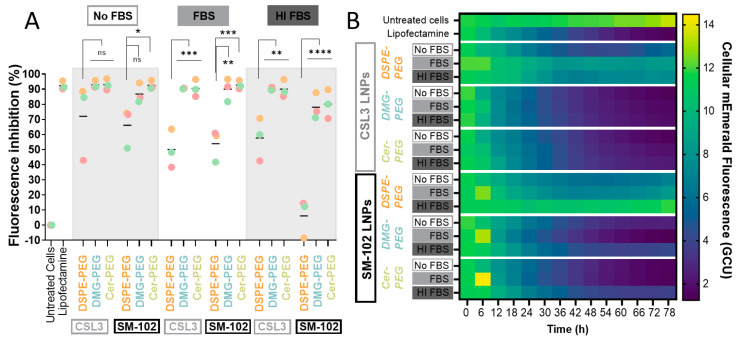
(**A**) mEmerald fluorescence inhibition of MDA-MB-231 cells transfected with CSL3 and SM-102 LNPs, including DSPE-, DMG- and Cer-PEG, in serum-free, FBS- or HI FBS-containing media 72 h after transfection. The mean fluorescence intensity of all of the cells of each condition were collected, and median values were calculated, normalized (%) to the control (untreated cells) and transformed in order to obtain the percentage of fluorescence inhibition. Each dot represents the result of one independent experiment (color-coded for experiments), and the horizontal bars represent the mean of three experiments. Statistical analyses were performed by two-way ANOVA followed by the Tukey test. (**B**) Heatmap showing the kinetic evolution of the mean fluorescence intensity (GCU) of MDA-MB-231 cells transfected with CSL3 and SM-102 LNPs including DSPE-, DMG- and Cer-PEG, over time in serum-free, FBS- or HI FBS-containing media. Each value represents the mean of three independent experiments (n = 3). The difference between groups was considered significant when the *p*-value was <0.05 (*), <0.01 (**), <0.001 (***) or <0.0001 (****).

## Data Availability

Not applicable.

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
