# Peer review of "Effect of PEG Anchor and Serum on Lipid Nanoparticles: Development of a Nanoparticles Tracking Method"

_pharmaceutics, 2023, doi:10.3390/pharmaceutics15020597_

Round 1

Reviewer 1 Report

The therapeutic LNPs are revisited  and their composition and behavior in serum are analyzed using the Nanoparticle Tracking Analysis approach and DLS. Serum protein adsorption on LNPs was ascribed to desorption of  shorter C14 lipid-PEG while longer  C18lipids –PEG are found more useful due to their stronger anchor to the NP.

Transfection efficacy in tumor cells was confirmed for ionizable lipids (CSL3)  as well as high siRNA encapsulation ability and lower toxicity. The authors find them promising for cargo delivery. The use of DSPE-PEG still remains unclear since longer PEG limit the transfection efficacy while shorter lipid-PEG do not eliminate interactions with proteins and formation of protein corona affecting  again transfection. This complicated and still unsolved obstacle is clearly described in the paper and the investigation is performed carefully so I suggest to accept the paper after the below mentioned suggestions are considered :

1.      1.  The discussion part is too long and can be easily shortened and made more consize.

2.       2. Page 16- line 50-52: If concentration of NPs is decreasing while size is increasing it means that aggregation of the NPs should be also considered not only protein adsorption following desorption of the lipid-PEG. The experiments done in pure buffer solution – without any proteins and in serum should presented in one figure or table and comment on the results of such comparison should be added.

Reviewer 2 Report

The present research manuscript titled “Effect of PEG anchor and serum on Lipid Nanoparticles: development of a nanoparticles tracking method” by Berger et al. is novel and exceptionally well written. The researchers conducted a series of in vitro experiments to prove their claim. The results are interesting and flow of the manuscript is excellent. However, I have some queries and the researchers are advised to address my comments. My comments are as follows.

Comment 1. Abstract: Kindly add concluding remark.

Comment 2. The density of the HepG2 and HeLa cells should be mentioned in cell culture studies.

Comment 3. The caption for Figure 3 is incomplete. Kindly revise.

Comment 4. The morphology of the developed nanoparticles should be analyzed and reported in the manuscript.

Comment 5. In my opinion, the researchers should conduct a short-term stability study and report in the manuscript.

Round 2

Reviewer 2 Report

The authors revised the manuscript very carefully as per my comments. I don't have further comments.